# Investigation of the Deterioration of Basu Granite Mechanical Properties Caused by Freeze–Thaw Cycles in High-Altitude Mountains in the Eastern Part of the Tibetan Plateau, China

Jixin Liu [1], Changbao Guo [1,2,*], Tianye Deng [1] and Sanshao Ren [1]

1   Institute of Geomechanics, Chinese Academy of Geological Sciences, Beijing 100081, China; liujixin@email.cugb.edu.cn (J.L.); 2102210035@email.cugb.edu.cn (T.D.); rensanshao123@163.com (S.R.)
2   Key Laboratory of Active Tectonics and Geological Safety, Ministry of Natural Resources, Beijing 100081, China
*   Correspondence: guochangbao@cags.ac.cn

**Abstract:** Mountains composed of granite are generally regarded as stable geological formations. However, in Alpine and high-altitude mountains in the eastern part of the Tibetan Plateau, geological hazards such as collapses and landslides occur frequently due to the deterioration of granite mechanical properties caused by the freeze–thaw cycles. To investigate this phenomenon, a freeze–thaw cyclic mechanical test is conducted on granite from the Basu area, and the rock's damage trend during the freeze–thaw process is analyzed through wave velocity and nuclear magnetic resonance (NMR) tests. The results indicated that the internal damage of granite increases and its wave velocity decreases significantly with increasing the freeze–thaw cycles, implying a decline in the rock's integrity. Furthermore, the development pattern of the NMR T2 relaxation time distribution indicates that the crack size range of naturally weathered rock samples further increased after freeze–thaw cycles, whereas less-weathered rocks showed a more concentrated range of crack sizes. Triaxial compression tests conducted on rock samples after the freeze–thaw cycles showed that parameters such as the uniaxial compressive strength, elastic modulus, internal friction angle, and cohesion of the rock decreased with increasing freeze–thaw cycles, while a significant change of Poisson's ratio was not observed. Based on the test data and theoretical analysis, a freeze–thaw damage constitutive model of the Basu granite can be established to simulate and predict the overall variation in rock stress and strain under various confining pressures and freeze–thaw cycles. Hopefully, the present study will provide useful guidance for research on the hazard mechanism and hazard prevention of granite sand-sliding slopes in the Basu area.

**Keywords:** Tibetan Plateau; granite; freeze–thaw cycles; sand-sliding slope; triaxial compression tests

## 1. Introduction

Mountains made up of granites are typically known for their stability, showcasing features such as low water absorption, high compressive strength, and durability. Natural granite cliffs can reach heights of more than 1000 m, such as the granite steep cliff on the top of the Circular Mound in Yosemite National Park in the United States which stands at 2698 m [1]. However, in alpine and high-altitude mountains of the eastern part of the Tibetan Plateau, geological hazards such as collapses, landslides, and sand-sliding slopes frequently occur due to the deteriorating mechanical properties of granite bodies caused by freeze–thaw cycles [2,3]. For instance, the Yigong landslide that occurred on 9 April 2000 in the granite body of the Zhamunong gully in Bomi County, Tibet had a maximum sliding distance of 6.7~7.0 km and a volume of over $3 \times 10^8$ m³ [4–7]. Some researchers have attributed the cause of such landslides to the intense freeze–thaw cycles of granite in areas with snow and ice [8].

Situated in the eastern part of the Tibetan Plateau, Basu County, located in the eastern part of the Tibetan Plateau, encounters a temperate semiarid climate which is a monsoon climate with a large diurnal temperature difference. Geological hazards, notably, sand-sliding slopes, often occur in granite slopes after freeze–thaw cycles [9,10], which have serious implications on the safe operation of Sichuan–Tibet Road G318. Temperature change is a significant factor in causing physical and chemical weathering of granite [11]. When the ambient temperature falls below 0 °C, the water in the rock freezes, prompting a water–ice phase transition, leading to the rock expanding and producing a frost heaving force, which results in new cracks that propagate all over the rock. As the temperature rises above 0 °C, the ice in the rock cracks melts, allowing water to infiltrate these cracks. During the subsequent freezing process, water refreezes, further amplifying the development of the cracks [12,13]. Microcracks and structural planes can be regarded as inherent defects within the rock. Experimental studies have additionally demonstrated that the stress and strain change processes of the rocks under an external load are essentially the processes of generating, expanding, and evolving new microcracks within the rock [9,14].

The distinctive cold environment of the plateau region gives rise to intricate damage properties of rocks during freeze–thaw cycles, garnering the interest of numerous researchers. Matsuoka [15] studied the freeze–thaw failure process of rocks in water through test analysis. Park et al. [16] conducted experiments to obtain the thermal conductivity, specific heat, and thermal expansion coefficient of rocks under different temperatures, providing a basis for studying the evolution law of the rock's mechanical properties under freeze–thaw cycles. Guo et al. [17] and Wang et al. [18,19] studied the mesoscopic damage extension and failure characteristics of rocks using Computed Tomography (CT), acoustic emission, and X-ray diffraction tests. They defined damage variables based on the CT number and obtained the law of damage development in the rock. Liu et al. [20] and Qiao et al. [21] studied rock damage tests under the coupling effect of freeze–thaw cycles and external loads. They demonstrated that rock damage is subject to a combination of freeze–thaw cycles and external loads and calculated the relationship between the crack tip strength factor and frost heaving force under frost heaving.

In the present paper, the freeze-weathering and sand-sliding slope of typical granite in the Basu area are investigated, and we further study the development process of freeze–thaw damage of granite by NMR tests, wave velocity tests, and triaxial compression mechanics tests. A freeze–thaw damage constitutive model is proposed, providing valuable insights into the formation mechanism and prevention of geological hazards in this area.

## 2. Geological Background

The Basu section of the Sichuan–Tibet Road G318 is situated in the central and eastern regions of the Tibetan Plateau, which is known for its complex geological structure. This area is marked by the distribution of the NNW-trending Nujiang fault zone and the nearly east–west Basu fault, along with their respective branch faults (Figure 1a). The exposed strata mainly consist of Jurassic and Cretaceous strata. Along the G318, the Nalong zone of Basu–Wangpai–Zaxize is mainly composed of the Cretaceous Moci Formation ($K_1 m$), Jida Town Formation ($K_1 j$), Tongkongnongba Formation ($K_2 t$), and $\gamma\pi$ granitic veins (Figure 1b). The granite in this area is mostly grayish white and light flesh-red, with a fine-grained to medium-grained structure. It is mainly composed of monzogranite and forms a geomorphic landscape characterized by high mountains, deep valleys, and steep slopes [22].

The national highway G318, stretching from Basu County to Ranwu, spans an average altitude of approximately 4000~4200 m. The regional climate is influenced by the altitude and exhibits an annual average temperature of approximately 9–10 °C, with the lowest temperatures reaching −20 °C. The average temperature in July is recorded at 19.2 °C, while the annual average precipitation is approximately 233 mm. Owing to the local climate and altitude, hillsides in the region are typically covered by glaciers, and frost weathering is particularly intense. The rock mass of the slopes undergoes seasonal freeze–thaw cycles over the years, resulting in the granite transforming into gravel and sand [23]. Gravel and

sand typically exhibit a rolling–sliding sand slope phenomenon under natural or external disturbances. In a field investigation, it was observed that slopes composed of granite along the Basu to Ranwu section of Sichuan–Tibet Road G318 are susceptible to a series of sand-sliding slope hazards [24]. For instance, sand-sliding slopes such as the one at Wangpai Hydropower Station on the left bank of Lengqu in Basu County (Figure 1b) and the large-scale sand-sliding slope at 3819 km of Sichuan–Tibet Road (Figure 1c) have been observed. These sand-sliding slopes continue to occur, posing severe threats to roads and human settlements [25].

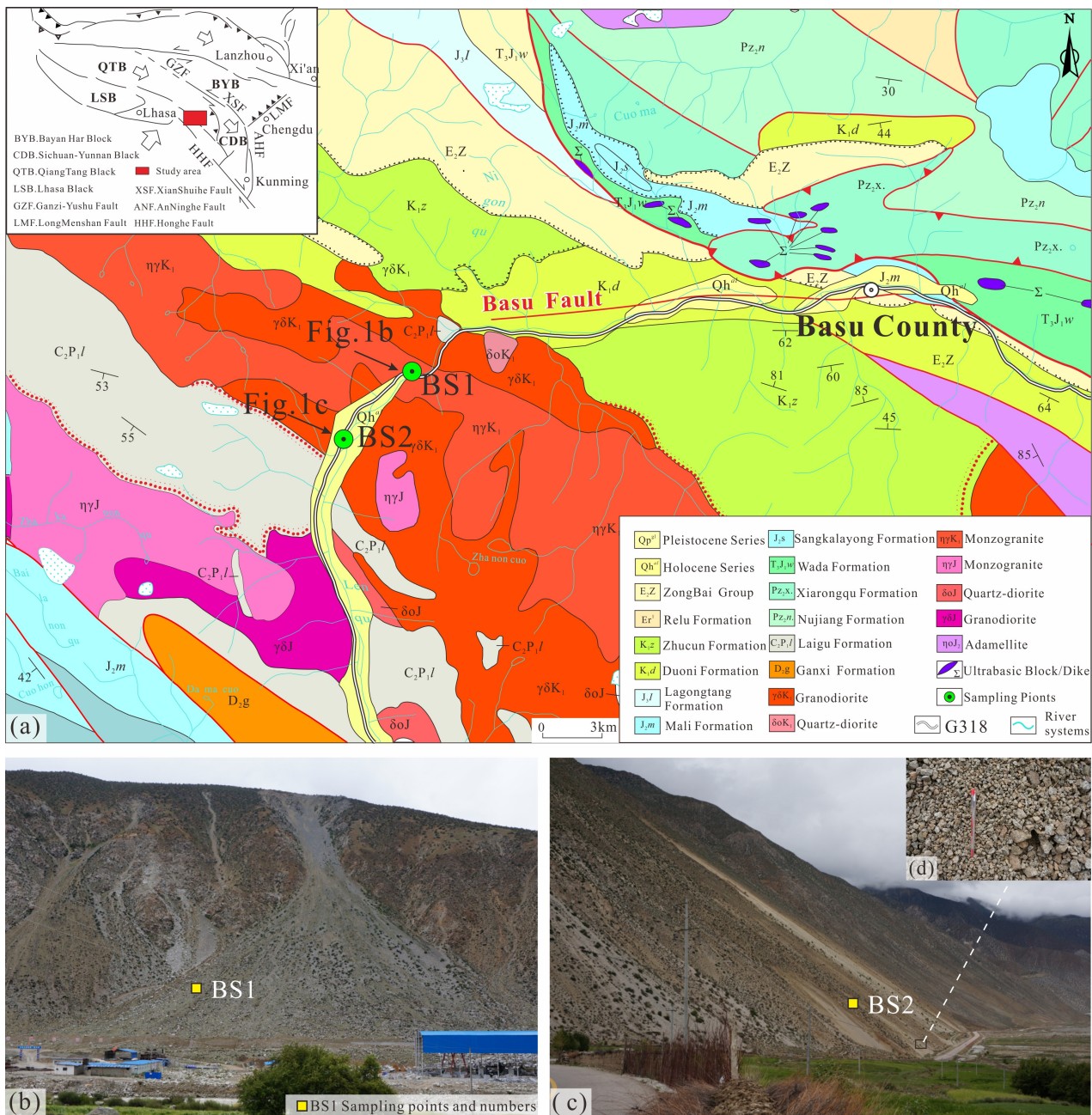

**Figure 1.** (**a**) Magmatic rock distribution area in the Palungzangbu Basin, southeast Tibet; (**b**) Sand-sliding slope of the WangPai Hydroelectric Station on the left bank of the LengQu River; (**c**) Sand-sliding slope hazard of the Sichuan–Tibet Road in the Basu area; (**d**) Granite gravel on the surface of the sand-sliding slope.

## 3. Methods

In order to investigate the internal damage evolution and mechanical strength variation of granite during freeze–thaw cycles, a comprehensive series of tests were conducted on the Basu granite including nuclear magnetic resonance (NMR) microstructure tests, water content tests, wave velocity tests, and triaxial compression mechanical tests. These tests were conducted over 0 to 40 freeze–thaw cycles and were based on basic physical property tests such as the specific gravity, dry density, water content, and porosity of the granite. The study analyzed the impact of freeze–thaw cycles on rock strength and deformation characteristics.

### 3.1. Sample Preparation

The granite samples utilized in this investigation were obtained from two locations, namely, Sichuan–Tibet Road at 3819 km in Basu County (BS1) and the sand-sliding slope of Wangpai Hydropower Station on the left bank of LengQu (BS2); both of them consist of monzogranite (Figure 1b,c). In accordance with the guidelines stipulated by the International Society of Rock Mechanics (ISRM), cylindrical specimens with a diameter of 50 mm and a height of 100 mm were extracted, ensuring that the nonparallelism between the upper and lower ends of each specimen was less than 0.02 mm. In total, 65 standard specimens were obtained (Table 1).

**Table 1.** Basic physical indices of the Basu granite samples used in the present study.

| Serial Number | Sample No. | Sampling Location | Sample Lithology | Number of Standard Samples (piece) | Specific Gravity | Dry Density (g/cm$^3$) | Water Content (%) | Porosity (%) |
|---|---|---|---|---|---|---|---|---|
| 1 | BS1-1 | Sand-sliding slope at 1 km north of Tongkong Village | monzogranite | 6 | 2.68 | 2.67 | 0.63 | 1.63 |
| 2 | BS1-2 | | | 4 | 2.75 | 2.64 | 0.65 | 1.78 |
| 3 | BS2-1 | Granite slope on the left bank of LengQu | monzogranite | 10 | 2.71 | 2.57 | 0.54 | 1.74 |
| 4 | BS2-2 | | | 10 | 2.64 | 2.58 | 0.76 | 1.73 |
| 5 | BS2-3 | | | 10 | 2.77 | 2.68 | 0.65 | 1.68 |
| 6 | BS2-4 | | | 25 | 2.72 | 2.55 | 0.64 | 1.98 |
| 7 | average | | | | 2.78 | 2.56 | 0.71 | 1.91 |

### 3.2. Test Scheme

#### 3.2.1. Freeze–Thaw Cycle Test

In this study, the direct freeze–thaw method was utilized to conduct tests on the Basu granite samples. The samples were categorized into 5 groups, with each group containing 9 samples. The freeze–thaw temperature was set at a range from −20 °C to 20 °C, and the rock samples were subjected to 0, 10, 20, 30, and 40 freeze–thaw cycles tests (Figure 2). The sample group subjected to 40 freeze–thaw cycles underwent mass and wave velocity tests at 0, 10, 20, 30, and 40 freeze–thaw cycles, and a scanning electron microscope (SEM) test was conducted for microscopic analysis.

The freeze–thaw testing procedure consisted of the following steps: (1) saturating the samples under vacuum, (2) freezing the samples at −20 °C ± 2 °C for 4 h, (3) thawing the samples in a constant temperature liquid bath at 20 °C for 4 h, and (4) repeating steps (1) to (3) multiple times to complete the required number of freeze–thaw cycles.

#### 3.2.2. Rock Mechanics Strength Test

The wave velocity tests were performed using an RSM wave velocity meter, while triaxial compression mechanical tests were conducted using the MTS815.03 rock triaxial test system at the Wuhan Institute of Geotechnical Mechanics, Chinese Academy of Sciences. For the triaxial compression tests, four groups of nine samples with the same number of

freeze–thaw cycles were prepared and tested under confining pressures of 0 MPa, 5 MPa, 10 MPa, and 15 MPa. The loading method was controlled by strain, and the loading rate was maintained at 0.001 mm/s. The freeze–thaw cycles and triaxial compression test scheme are presented in Table 2.

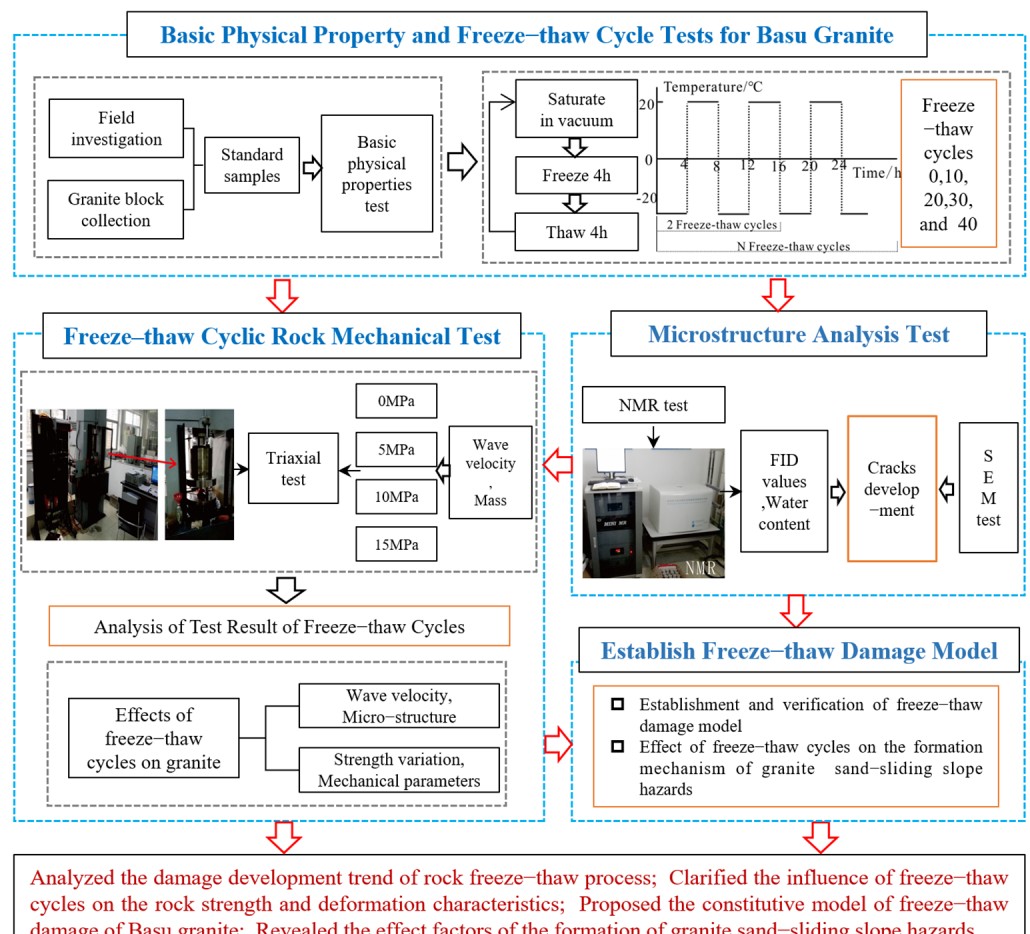

**Figure 2.** Scheme of the technical work of the present study.

**Table 2.** Test scheme of freeze–thaw cycles and triaxial compression for the Basu granite.

| Group No. | Sample No. | Freeze–Thaw Cycles | Confining Pressure $\sigma_3$ (MPa) | Group No. | Sample No. | Freeze–Thaw Cycles | Confining Pressure $\sigma_3$ (MPa) |
|---|---|---|---|---|---|---|---|
| Group 1 | BS2-2-2 | 0 | 0 | Group 2 | BS2-1-8 | 10 | 0 |
| | BS2-2-3 | | | | BS2-1-10 | | |
| | BS2-1-1 | 0 | 5 | | BS2-3-1 | 10 | 5 |
| | BS2-1-3 | | | | BS2-3-2 | | |
| | BS2-1-4 | 0 | 10 | | BS2-3-3 | 10 | 10 |
| | BS2-1-5 | | | | BS2-3-4 | | |
| | BS2-1-6 | 0 | 15 | | BS2-3-5 | 10 | 15 |
| | BS2-1-7 | | | | BS2-3-6 | | |

**Table 2.** *Cont.*

| Group No. | Sample No. | Freeze–Thaw Cycles | Confining Pressure $\sigma_3$ (MPa) | Group No. | Sample No. | Freeze–Thaw Cycles | Confining Pressure $\sigma_3$ (MPa) |
|---|---|---|---|---|---|---|---|
| Group 3 | BS2-3-7 | 20 | 0 | Group 4 | BS2-4-14 | 30 | 0 |
| | BS2-3-8 | | | | BS2-4-15 | | |
| | BS2-3-10 | 20 | 5 | | BS2-4-17 | 30 | 5 |
| | BS2-4-9 | | | | BS2-4-18 | | |
| | BS2-4-10 | 20 | 10 | | BS2-4-19 | 30 | 10 |
| | BS2-4-11 | | | | BS2-4-20 | | |
| | BS2-4-12 | 20 | 15 | | BS2-4-21 | 30 | 15 |
| | BS2-4-13 | | | | BS2-4-22 | | |
| Group 5 (Wave Velocity, NMR tests) | BS2-4-23 | 40 | 0 | Group 5 (Wave Velocity, NMR tests) | BS2-4-5 | 40 | 10 |
| | BS2-4-25 | | | | BS2-4-6 | | |
| | BS2-4-3 | 40 | 5 | | BS2-4-7 | 40 | 15 |
| | BS2-4-4 | | | | BS2-4-8 | | |

## 4. Analysis of Results and Discussion

### 4.1. Effect of Freeze–Thaw Weathering on Wave Velocity

The propagation of microcracks in rock is a crucial process in the development of internal damage [26]. Microcracks can hinder the transmission of sound waves, therefore influencing wave velocity measurements [27]. Wave velocity tests were conducted on the Basu granite samples subjected to different numbers of freeze–thaw cycles, revealing a decrease in wave velocity from the original range of 5000~5500 m/s to 4000~5000 m/s, indicating a progression of internal damage in the rock due to the number of freeze–thaw cycles [28]. The trend of decreasing wave velocity with increasing number of freeze–thaw cycles (Figure 3) is indicative of a corresponding enhancement in the microcrack development within the granite samples.

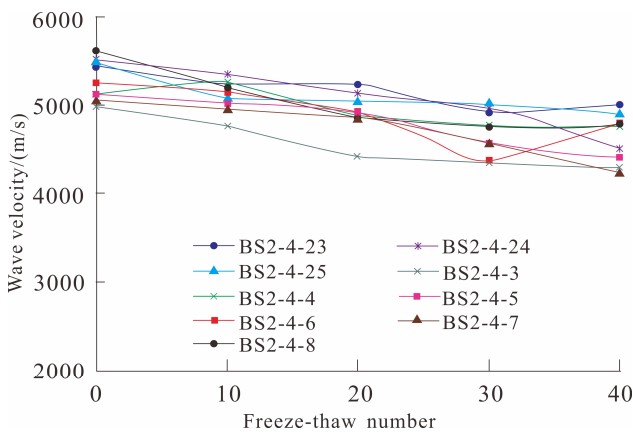

**Figure 3.** The variation curves of wave velocity with the number of freeze–thaw cycles of the Basu granite.

### 4.2. Effect of Freeze–Thaw Weathering on Rock Microstructure Deterioration

To explore the impact of freeze–thaw cycles on the deterioration of granite structures, nuclear magnetic resonance (NMR) technology was employed to analyze the pore structure. The NMR method is a nondestructive testing technique that enables rapid and convenient analysis of the water content and energy state of a porous medium by measuring the

response of polar molecules, such as water molecules, under a magnetic field [29]. The PQ-001 MiniNMR low-field nuclear magnetic resonance testing system of the Wuhan Institute of Rock and Soil Mechanics, Chinese Academy of Sciences was utilized for the NMR test. During the test, the NMR signal typically displays an exponential free decay known as free-induction decay (FID) [30,31].

The initial value of the FID curve, referred to as the FID peak value, offers an indirect measure of water content [32]. However, the FID peak value is affected by the temperature and magnetic mineral content, which can be mitigated by testing the same sample at different water contents and temperatures. By establishing a proportional relationship between the FID peak values and the water content, subsequent FID peaks can then be used to estimate the water content [33]. The calculation (Equation (1)) is:

$$w_n = w_1 + \frac{w_2 - w_1}{F_2 - F_1}(F_n - F_1) \tag{1}$$

where, $w_1$, $w_2$, and $w_n$ are the water contents measured of the first, second, and $n$th measurements, respectively; and $F_1$, $F_2$, and $F_n$ are the FID peak values of the first, second, and $n$th measurements, respectively.

Based on the FID peak diagram of the nuclear magnetic signals obtained after subjecting the Basu granite to different freeze–thaw cycles (Figure 4), it can be observed that the FID peak values varied significantly with each freeze–thaw cycle. The distribution of the water content diagram of the Basu granite under different freeze–thaw cycles can be calculated using Equation (1) (Figure 5). The water content exhibits a slight decrease with an increase in the number of freeze–thaw cycles, aligning with the observed trend in density tests after freeze–thaw cycles (the right axis of Figure 5). This trend may be attributed to the fact that the rock damage caused by freeze–thaw cycles primarily comprises a few small cracks. Given the small volume of these cracks, the amount of water entering them during the saturation process of the rock sample under vacuum is limited. Additionally, the partial disintegration of the rock samples due to freeze–thaw damage could potentially offset the increase in mass caused by water [34,35].

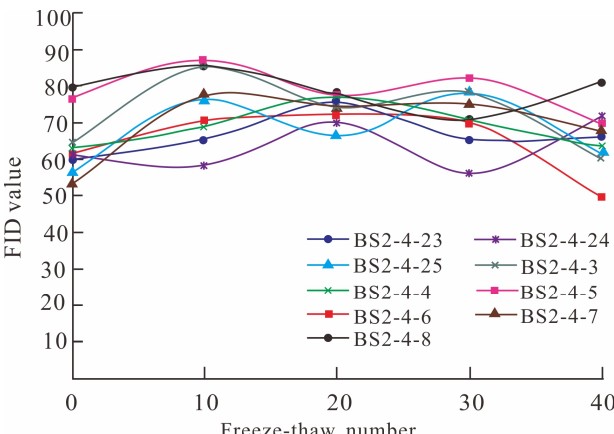

**Figure 4.** The FID values of the Basu granite after the freeze–thaw cycles.

NMR is a technique that characterizes the quality of a rock mass by analyzing the relaxation process of fluid within its pores. The T2 relaxation time, which describes the time required for water molecules to return to their original state after being stimulated by a magnetic field [36], reflects the binding effect of the rock pores on water. The T2 value correlates with the size of the pores: smaller pores impose a more significant binding effect on the water molecules, resulting in a smaller T2 value, and vice versa.

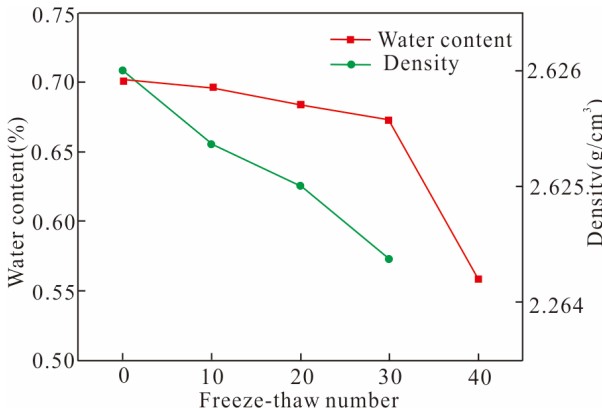

**Figure 5.** The curves of water content and density with the number of freeze–thaw cycles.

The T2 distribution map derived from the typical Basu granite rock samples (Figure 6) reveals that a single peak shape is observed after undergoing freeze–thaw cycles. This indicates a relatively uniform distribution of crack sizes within the samples. Since the rock samples are relatively intact prior to the freeze–thaw cycle, the T2 peak values primarily represent cracks that arise as a result of the cycle. Furthermore, the T2 values of the Basu granite samples, ranging from 3 to 200 ms, display significant fluctuations due to the freeze–thaw cycles.

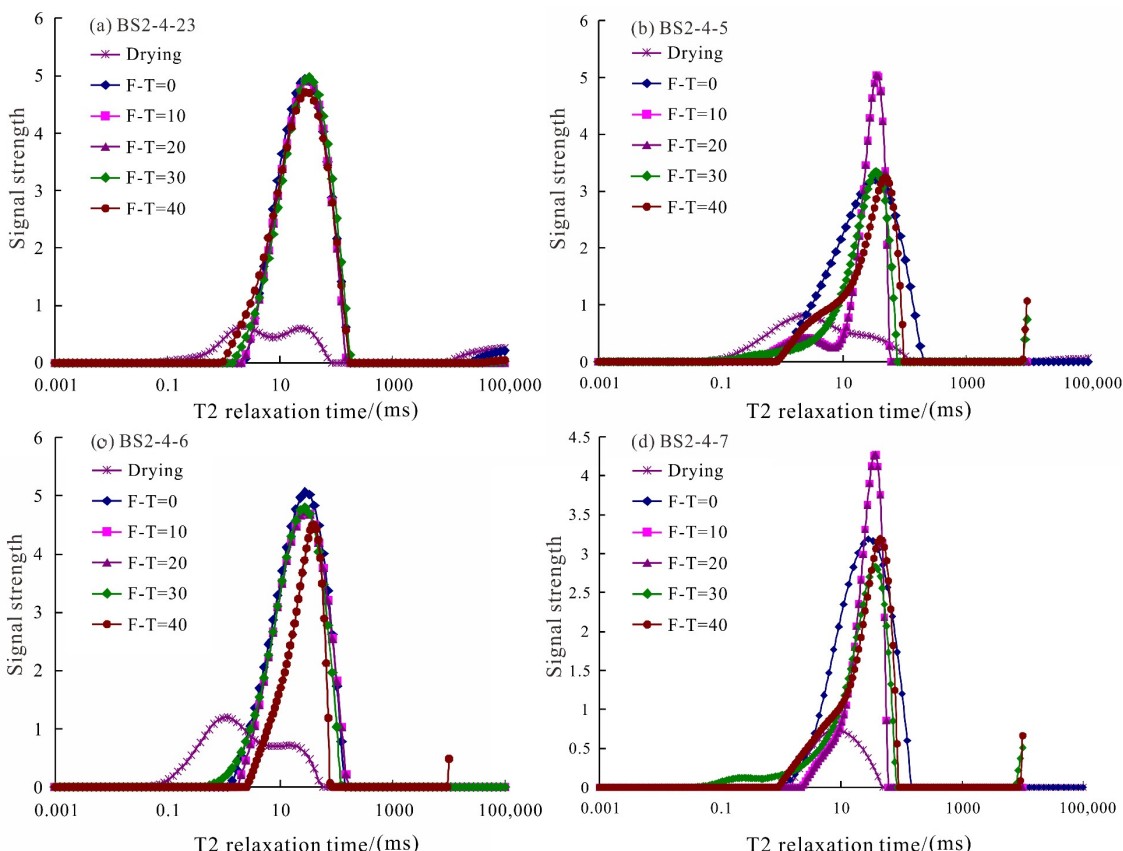

**Figure 6.** The curves of T2 distribution for the representative testing samples (**a**) Sample BS-2-4-23; (**b**) Sample BS-2-4-5; (**c**) Sample BS-2-4-6; (**d**) Sample BS-2-4-7.

The scanning electron microscopy results of granite after undergoing freeze–thaw weathering (Figure 7) demonstrate the formation of cracks within the sample, primarily ranging from 5 to 30 μm in size, under the influence of frost heaving forces. This phe-

nomenon can be attributed to the stress concentration resulting from uneven expansion and contraction of the rock minerals during the freeze–thaw cycles. Simultaneously, water frost-heave in the cracks could have prompted crack enlargement and connectivity, with most cracks propagating along the boundaries of granite grains [37].

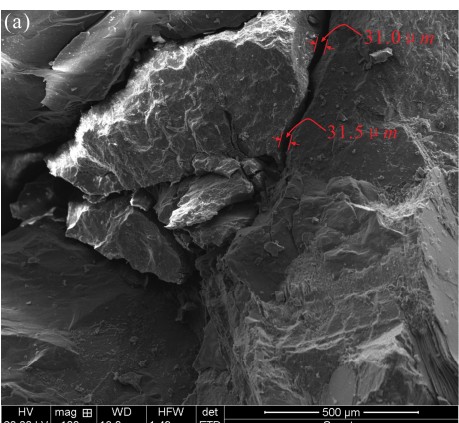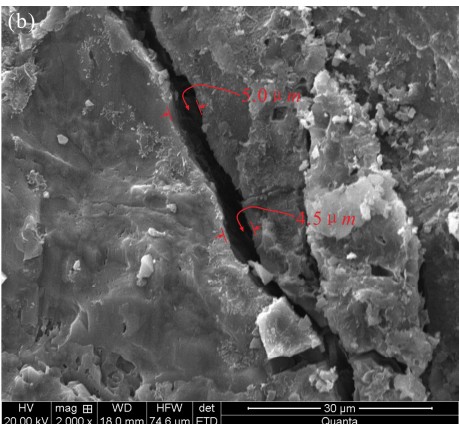

**Figure 7.** Typical SEM images of the Basu granite after freeze–thaw cycles (**a**) Expansion of cracks; (**b**) Generation of new micro-cracks.

*4.3. Stress–Strain Analysis of the Rock Triaxial Compression Test*

4.3.1. Volume Variation of Typical Rock Samples

The stress–strain behavior of the Basu granite samples displays an initial slow increase in stress with strain during the early stage of the test (Figure 8b), accompanied by a volume strain increase that is indicative of axial strain, indicating sample volume reduction (Figure 8c). This phenomenon primarily results from crack compaction due to axial forces [38]. Upon further loading, the axial stress growth rate accelerates with the strain, leading to an approximately linear stress–strain relationship. The rock sample then enters the stage of elastic deformation. Following elastic deformation, the axial stress growth rate decreases with strain, and the volume strain decreases with increasing axial strain. At this stage, the rock sample experiences volume expansion (Figures 8 and 9). Upon entering the plastic stage, the rock is loaded at a constant strain rate, and the stress–strain curve reaches the peak point, signifying the attainment of the strength value. Beyond the peak point, the axial stress rapidly decreases, accompanied by rapid volume strain reduction. The sample volume then experiences a violent expansion, indicating a macroscopic fracture of the rock (Figure 8a).

4.3.2. Characteristics of Triaxial Compression Deformation

Based on the stress–strain curves of granite under different confining pressures and freeze–thaw cycles (Figure 9), it is evident that the stress–strain curves of granite before and after freeze–thaw cycles exhibit a similar shape. The curve can be categorized into compaction, elastic deformation, plastic deformation, and post-peak failure stages. The interior of the rock samples was adversely impacted by the thermal expansion, contraction deformation, and frost heaving force, causing the development and penetration of internal cracks, which led to a decline in the strength of the granite samples. The peak strength and post-peak residual strength of the rock samples display a gradually declining trend [39,40]. For instance, at a confining pressure of 5 MPa, the peak axial stress of the nonfreeze-thaw granite is 187.19 Mpa (Figure 9b). In contrast, the peak axial stresses after 10, 20, 30, and 40 freeze–thaw cycles are 170.16 MPa, 172.55 MPa, 149.27 MPa, and 139.07 MPa, respectively, signifying decreases of 9.1%, 7.8%, 20.3%, and 25.7%.

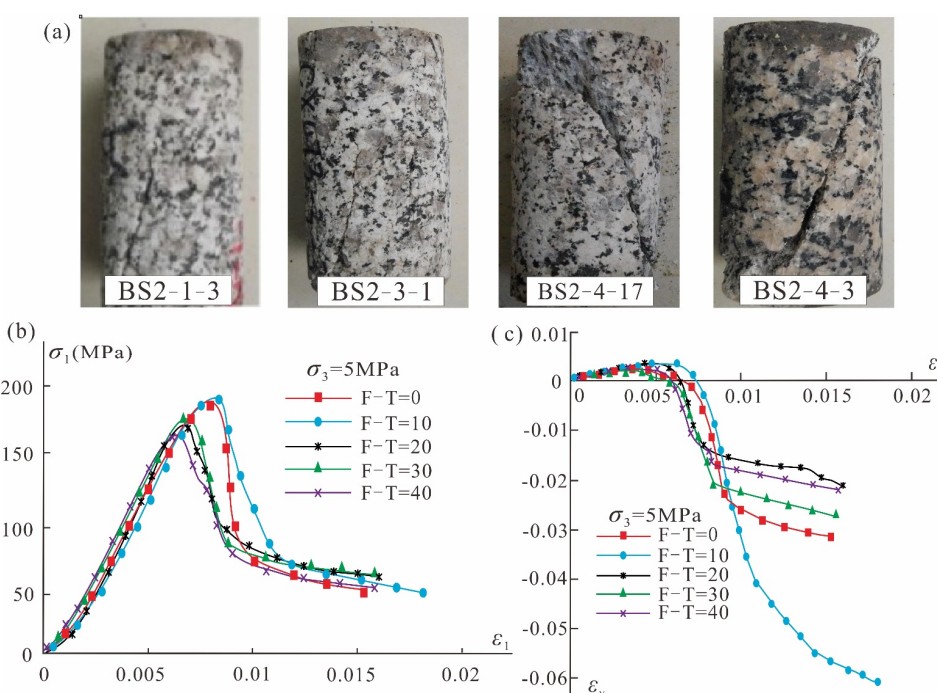

**Figure 8.** (**a**) Macro-fracture of rocks under different numbers of freeze–thaw cycles; (**b**) the curves of stress–strain for typical samples; (**c**) the curves of volume strain–axial strain for typical samples.

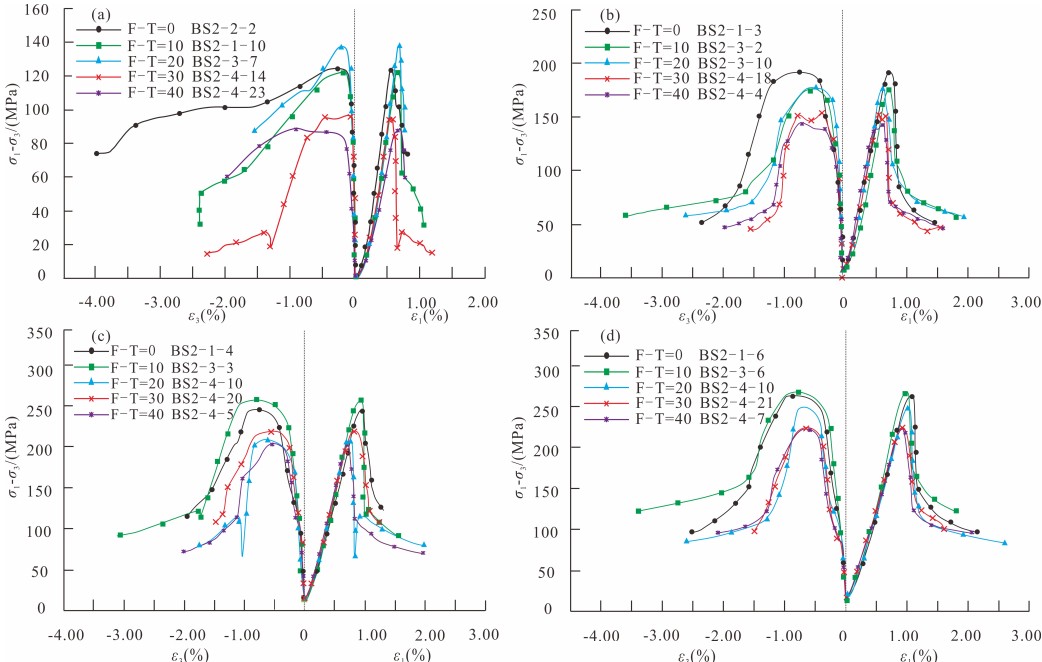

**Figure 9.** The curves of stress–strain with different numbers of freeze–thaw cycles for the Basu granite. (**a**) $\sigma_3 = 0$ MPa; (**b**) $\sigma_3 = 5$ MPa; (**c**) $\sigma_3 = 10$ MPa; (**d**) $\sigma_3 = 15$ MPa.

The stress–strain curves of granite display overlapping compaction stages, while the freeze–thaw cycles result in shortened stages of elastic and plastic deformation. An increase in the number of freeze–thaw cycles leads to a brittle failure tendency of rock samples. The reduction in the slopes of stress–strain curves with increasing freeze–thaw cycles indicates the decrease in the elastic modulus of the rock samples. Furthermore, as the number of freeze–thaw cycles increases, the rock samples display brittle failure at earlier stages. With the increasing confining pressure, the stress reduction rate during the post-peak stage of

the stress–strain curve exhibits a declining trend, while the residual strength continues to increase. Moreover, the post-peak yield platform gradually becomes more evident, and the rock samples transform from brittle failure to plastic failure.

### 4.3.3. Granite Mechanical Strength

In order to conduct a more comprehensive analysis of the mechanical characteristics of the Basu granite, the peak axial stress in the stress–strain curve was designated as the rock's peak strength. The relationship between peak strength and confining pressure was established based on the test results of each group, resulting in a $\sigma_1$-$\sigma_3$ scatter diagram. To determine the strength parameters internal friction angle $\varphi$ and cohesion $c$, the data points were subjected to curve-fitting based on a linear function to obtain the slope $k$ and intercept $b$. Subsequently, $k$ and $b$ were converted into the aforementioned strength parameters by means of Mohr–Coulomb's law, as detailed in Equations (2) and (3):

$$\varphi = 2\left(\cot \sqrt{k} - \frac{\pi}{2}\right) \tag{2}$$

$$c = \frac{b}{2\sqrt{k}} \tag{3}$$

For additional investigation of the mechanical properties of the Basu granite, the peak axial stress from the stress–strain curves was utilized as the rock's peak strength. By examining representative samples of the Basu granite subjected to 0, 10, 20, 30, and 40 freeze–thaw cycles, the variation of strength with confining pressure was obtained (Figure 10a,c,e,g,i). The strength variation exhibited a notably linear relationship with the confining pressure. The corresponding slope $k$ and intercept $b$ were then obtained through linear fitting, and the values of the internal friction angle and cohesion were subsequently calculated via Equations (2) and (3), respectively.

The resulting calculations revealed that the internal friction angle and cohesion were 57.30° and 16.82 MPa after 0 freeze–thaw cycles, 58.39° and 16.67 MPa after 10 freeze–thaw cycles, 52.99° and 22.33 MPa after 20 freeze–thaw cycles, 54.13° and 15.96 MPa after 30 freeze–thaw cycles, and 54.82° and 14.12 MPa after 40 freeze–thaw cycles. Subsequently, Figure 10b,d,f,h,j demonstrate the corresponding stress Mohr's circles and strength envelopes for the respective failure states.

### 4.4. Analysis of the Effect of Freeze–Thaw Cycles on Rock Mechanical Parameters

In rock mechanics testing, the portion of the stress–strain curve that exhibits a linear relationship before the volume strain reaches its maximum value is commonly referred to as the elastic deformation stage. The elastic modulus of the material can be calculated from the stress–strain curve in this stage for a uniaxial test, while the Poisson's ratio can be determined by analyzing the lateral strain–axial strain curve in the same stage. The variations in the elastic modulus and Poisson's ratio of the Basu granite with respect to the number of freeze–thaw cycles are illustrated in Figure 11a,b, respectively.

As depicted in Figure 11a, the uniaxial elastic modulus of the Basu granite samples ranges from 20 to 35 GPa and displays a noticeable reduction with increasing numbers of freeze–thaw cycles. Conversely, Figure 11b shows that Poisson's ratio does not follow a clear trend with respect to the number of freeze–thaw cycles, and some of the results exhibit substantial variability. The mean Poisson's ratio of the Basu granite samples is 0.31.

Following triaxial mechanical tests on the Basu granite specimens exposed to various freeze–thaw cycles, the shear strength parameters of the granite, namely, the internal friction angle and cohesion, were determined (Table 3). In general, both the internal friction angle and cohesion decrease with increasing numbers of freeze–thaw cycles, although their trends differ slightly. The cohesion of the Basu granite specimens exhibits little variation, while the internal friction angle displays a significant reduction.

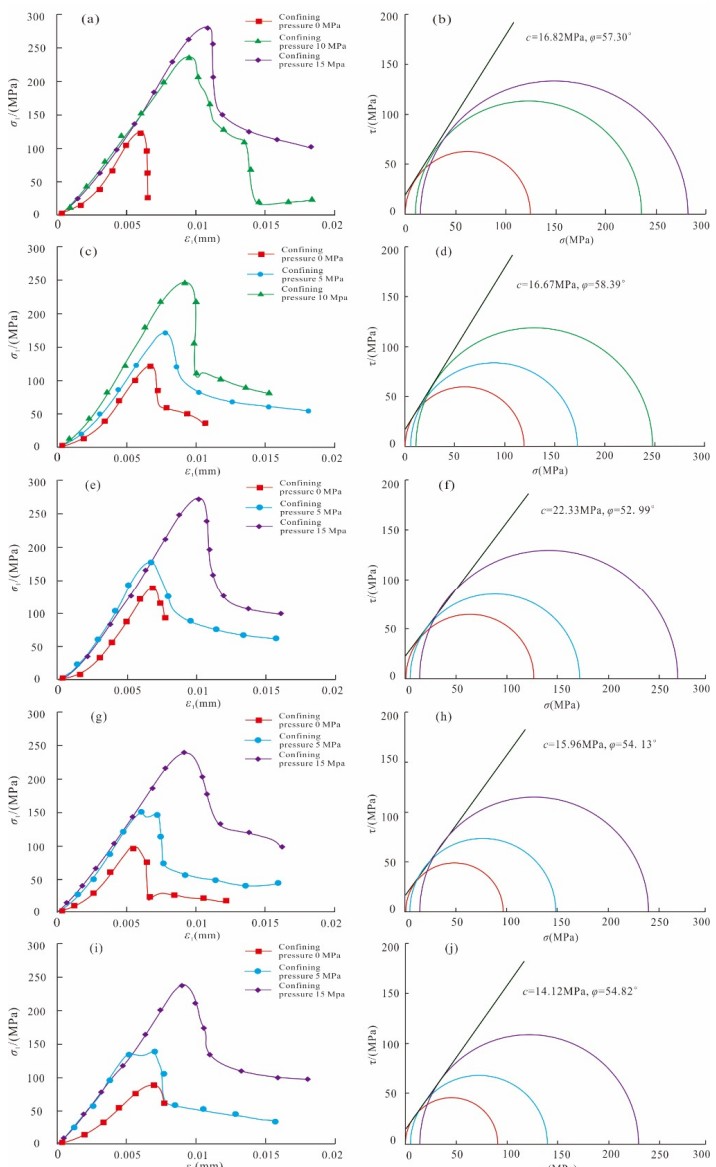

**Figure 10.** The variation in the sample strength with the confining pressure, Mohr's circle, and the strength envelope line with different numbers of freeze–thaw cycles. (**a**,**c**,**e**,**g**,**i**) show the changes in sample strength with different confining pressures at 0, 10, 20, 30, and 40 freeze–thaw cycles; (**b**,**d**,**f**,**h**,**j**) show the stress Mohr's circles and the strength envelopes of the samples at 0, 10, 20, 30, and 40 freeze–thaw cycles.

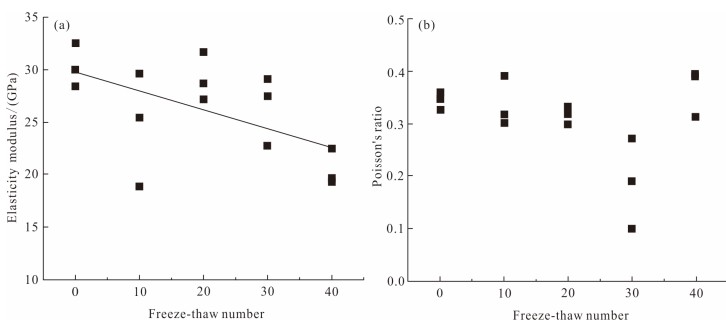

**Figure 11.** (**a**) The plot of variation in the elasticity modulus with freeze–thaw cycles; (**b**) the plot of granite the Poisson's ratio with freeze–thaw cycles.

**Table 3.** The variation in typical mechanical strength parameters of granite with the number of freeze–thaw cycles.

| Freeze–Thaw Cycles | Sample No. | Confining Pressure $\sigma_3$ (MPa) | Peak Axial Stress (MPa) | Internal Friction Angle $\varphi$ (°) | Cohesion $c$ (MPa) | Elastic Modulus $E$ (GPa) | Poisson's Ratio $\mu$ |
|---|---|---|---|---|---|---|---|
| 0 | BS2-2-3 | 0 | 124.00 | 57.30 | 16.82 | 30.005 | 0.3486 |
| | BS2-1-4 | 5 | 236.79 | | | | |
| | BS2-1-6 | 10 | 281.04 | | | | |
| 10 | BS2-1-10 | 0 | 122.77 | 58.39 | 16.67 | 29.624 | 0.3193 |
| | BS2-3-2 | 10 | 170.16 | | | | |
| | BS2-3-3 | 15 | 247.56 | | | | |
| 20 | BS2-3-7 | 0 | 137.16 | 52.99 | 22.33 | 31.696 | 0.3328 |
| | BS2-3-10 | 5 | 172.55 | | | | |
| | BS2-4-12 | 10 | 269.23 | | | | |
| 30 | BS2-4-14 | 0 | 96.64 | 54.13 | 15.96 | 27.479 | 0.2745 |
| | BS2-4-18 | 5 | 149.29 | | | | |
| | BS2-4-21 | 10 | 240.80 | | | | |
| 40 | BS2-4-23 | 0 | 88.93 | 54.82 | 14.12 | 19.299 | 0.3963 |
| | BS2-4-4 | 5 | 139.07 | | | | |
| | BS2-4-7 | 10 | 238.22 | | | | |

The peak strength variation of the Basu granite samples with the number of freeze–thaw cycles (Figure 12) indicates a decrease in rock strength with an increase in the number of freeze–thaw cycles under different confining pressures. This trend follows a near-linear pattern. Specifically, under the uniaxial test condition where the confining pressure is zero, the attenuation of rock strength due to freeze–thaw cycles is minimal. Conversely, at higher confining pressures, the impact of freeze–thaw cycles on the degree of attenuation of rock strength is more significant.

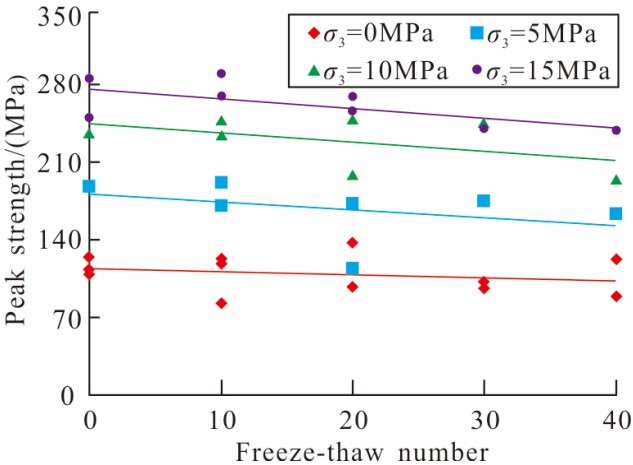

**Figure 12.** Variation in the peak strength with the number of freeze–thaw cycles.

*4.5. Constitutive Model of Freeze–Thaw Damage of the Basu Granite*

4.5.1. Improved Expression of Rock Damage Variable

In accordance with the principles of damage mechanics, it is understood that the material has already incurred damage when rocks transition into the elastic–plastic deformation

stage. This implies that the process of plastic deformation is synonymous with the gradual increase in the damage factor. Rabotnov [36] introduced the concept of a continuously distributed internal variable termed as the damage factor, denoted by $D$, and put forth its definition:

$$D = \frac{\left(A - \tilde{A}\right)}{A} \tag{4}$$

where $D$ is the damage factor, $A$ is the total area of the material, and $\tilde{A}$ is the effective area of the material that has not been damaged and can bear the load. The magnitude of the damage factor is related to the severity of the rock damage, and the damage factor always ranges from 0 to 1.

In order to facilitate the measurement of rock damage, Lemaitre [41] introduced the concept of effective stress. It is assumed that the material follows the generalized Hooke's law before the onset of damage and that there is no lateral damage [42], then:

$$\sigma_1 = (1 - D)[E\varepsilon_1 + \mu(\sigma_2 + \sigma_3)] + D\sigma_R \tag{5}$$

where $E$ is the elastic modulus; $\mu$ is Poisson's ratio; $\sigma_1$, $\sigma_2$, and $\sigma_3$ are the maximum principal stress, intermediate principal stress, and minimum principal stress, respectively; and $\sigma_R$ is the residual strength.

As illustrated in Figure 8b, the stress gradually increases with strain due to the compaction of pre-existing cracks in the material prior to elastic deformation. During this stage, no new damage occurs, and the original damage is actually mitigated by compaction [43,44]. While this study exclusively focuses on damage evolution during compression, it does not account for any damage resulting from rock cracks prior to compression. Hence, it is imperative to address the compaction stage of the nonlinear elastic phase. To this end, a reverse extension line is drawn along the elastic straight line of the stress–strain curve, and the intersection with the horizontal axis is designated as $\varepsilon_p$. The original coordinate system is then shifted to the right by $\varepsilon_p$ to establish a new coordinate system (Figure 13). Then, the new axial strain $\varepsilon_1'$ is expressed as:

$$\varepsilon_1{}' = \varepsilon_1 - \varepsilon_p \tag{6}$$

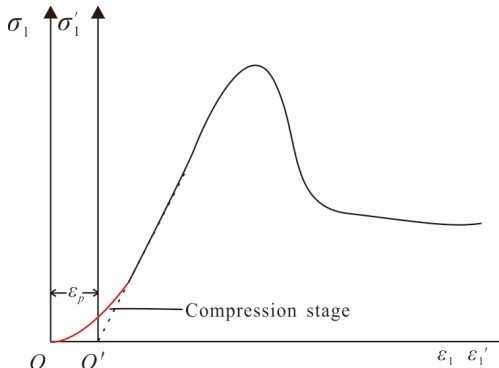

**Figure 13.** The variation of stress–strain in the new and old coordinate systems.

The damage factor expression in the new coordinate system is:

$$D = \frac{\sigma_1 - E\varepsilon_1{}' - \mu(\sigma_2 + \sigma_3)}{\sigma_R - E\varepsilon_1{}' - \mu(\sigma_2 + \sigma_3)} \tag{7}$$

In the elastic stage, according to Hooke's law ($\sigma_1 - E\varepsilon_1' - \mu(\sigma_2 + \sigma_3) = 0$), the damage factor $D = 0$ can be obtained from Equation (7). In the initial stage of the new coordinate system, the stress–strain curve is still an elastic straight line higher than the origin (Figure 13). At this stage, the material is transitioning from compaction to elastic deformation. As the

stress is greater than the elastic stress, $\sigma_1 - E\varepsilon_1' - \mu(\sigma_2 + \sigma_3) > 0$, and $D > 0$. However, the damage factor evolution curve shows abnormal behavior (Figure 14), which is inconsistent with the real situation. Therefore, the data in the abnormal damage stage should be eliminated when the damage factor is measured.

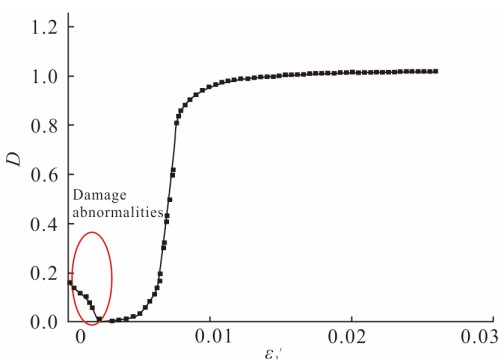

**Figure 14.** Simplification of the damage factor variation.

From Figure 14, it is observed that the damage factor is very small in the elastic stage and begins to increase sharply when the sample enters plastic deformation. When the rock strength curve crosses the peak point, the growth rate of the damage factor decreases, and it gradually approaches one, indicating complete failure of the rock. In general, the damage evolution curve follows an 'S' shape, and the damage evolution equation should reflect the gentle section after the damage gradually stabilizes to reflect the stress–strain relationship before and after rock failure. Liu et al. [42] used Equation (8) to calculate the development process of the damage factor:

$$D = \frac{1}{1 + e^{a - r\varepsilon_1'}} \tag{8}$$

where $a$ and $r$ are material parameters. $a$ reflects the initial damage degree, and $r$ reflects the damage growth rate.

### 4.5.2. Constitutive Model of Freeze–Thaw Damage

To establish a more comprehensive constitutive model, it is essential to consider the significant effects of confining pressure and freeze–thaw cycles on rock strength [45]. The constitutive equation parameters, as given in Equation (5), include $E$, $\mu$, $D$, and $\sigma_R$. Once the material parameters are determined, the damage factor evolution process and the entire stress–strain curve can be obtained. Generally, the material parameters are related to the confining pressure and the internal structural state of the material. Due to freeze–thaw cycles, microcracks inside the rock increase, leading to a change in the internal structural state of the rock. As a result, the four material parameters show some variations under different confining pressures and freeze–thaw cycles. As depicted in Figure 11, except for a slightly lower value of $E$ observed for 40 freeze–thaw cycles, the other parameter values are relatively stable, while the values of $\mu$ show some variation. Therefore, in this study, the material parameters $E$ and $\mu$ are regarded as constants. Through an analysis of the damage factor $D$ and residual strength $\sigma_R$, the relationship between them and confining pressure and freeze–thaw cycles is established, and a freeze–thaw damage constitutive equation that can describe the entire process of stress and strain is proposed.

In the triaxial tests, the axial stress of the rock undergoes a sharp decline after reaching the peak strength and subsequently declines gradually until it stabilizes at a residual strength value. For instance, in the case of the Basu granite, the rock stress–strain curve shows little variation in stress after a sharp attenuation stage when the axial strain reaches 0.015. The stress value corresponding to this strain is taken as the residual strength. The test data for the rock samples with different confining pressures and freeze–thaw cycles were analyzed to obtain the variation curve of residual strength with freeze–thaw cycles,

as shown in Figure 15. It is evident from the figure that the residual strength generally increases with increasing confining pressure and decreases with increasing freeze–thaw cycles. A regular pattern shown in Equation (9) is obtained by regression analysis:

$$\sigma_R = c_1 N + c_2 \sigma_3 + c_3 \tag{9}$$

where $N$ and $\sigma_3$ represent the number of freeze–thaw cycles and the confining pressure, respectively; and $c_1$, $c_2$, and $c_3$ are test parameters.

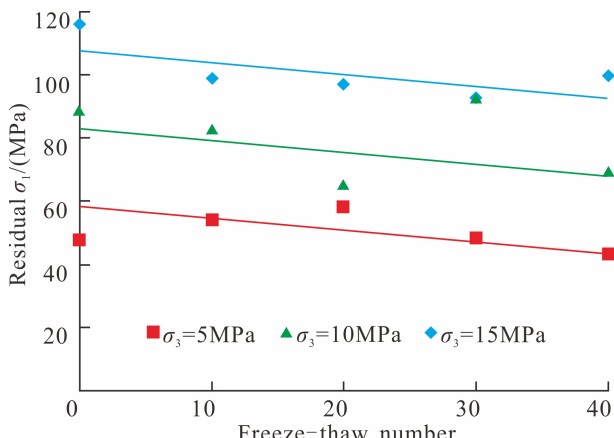

**Figure 15.** Variation in residual strength with the number of freeze–thaw cycles.

Equation (9) presents a linear relationship between the residual strength and the confining pressure as well as the number of freeze–thaw cycles. As demonstrated in Figure 15, the calculated results are compared with the experimental values using a solid line, indicating that Equation (9) is an effective model for describing the variation in residual strength with confining pressure and freeze–thaw cycles. The calculated values for the material parameters in Figure 15 are $c_1 = -0.384$ MPa, $c_2 = 4.938$ MPa, and $c_3 = 33.93$ MPa.

As per Equation (8), the damage factor $D$ includes two parameters: the initial damage degree $a$ and the damage growth rate $r$. Therefore, it is necessary to analyze these two parameters separately. By performing mathematical transformations on Equation (8), we can obtain:

$$ln\left(\frac{1}{D} - 1\right) = a - r\varepsilon_1' \tag{10}$$

The aforementioned equation demonstrates that $ln(^1/_D - 1)$ has a linear relationship with the axial strain $\varepsilon_1'$ in the new coordinates, with a slope of $-r$ and an intercept of $a$. Thus, to obtain the relationship between $ln(^1/_D - 1)$ and $\varepsilon_1'$, the damage factors acquired from the tests were measured. However, due to the abrupt decrease in axial stress after reaching peak strength during the test and the misalignment of some samples after rupture, the stress–strain curve becomes discontinuous. Consequently, the regularity of the damage factors is poor at this stage. Therefore, the data considered for the parametric analysis of the damage factor included only the elastic deformation stage up to the rapid decrease in the axial stress. These data capture the main characteristics of rock deformation and failure, such as the generation and development of material plastic deformation and the fluctuation of the stress peak. After sorting, the relationship between $ln(^1/_D - 1)$ and $\varepsilon_1'$ under different confining pressures and freeze–thaw cycles was obtained (Figure 16).

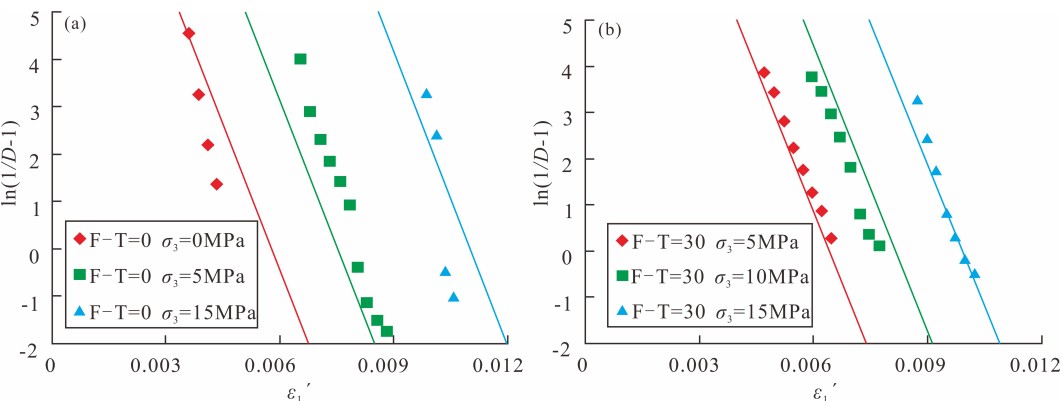

**Figure 16.** The variation of $ln(^1/_D - 1)$ with $\varepsilon_1'$ (**a**) F–T = 0; (**b**) F–T = 30.

The plot in Figure 16 illustrates that $ln(^1/_D - 1)$ exhibits a linear relationship with $\varepsilon_1'$ under varying confining pressures and freeze–thaw cycles, with similar slopes but different intercepts. Thus, the damage growth rate $r$ may be considered a constant, and the initial damage degree $a$ can be represented as a function of the confining pressure and the number of freeze–thaw cycles. In this study, Equation (11) was employed:

$$a = d_1 N + d_2 \sigma_3 + d_3 \tag{11}$$

The above equation demonstrates that the initial damage degree $a$ has a linear relationship with both the number of freeze–thaw cycles and the confining pressure.

In order to obtain the parameters for the model, Equation (11) was substituted into Equation (10), followed by linear regression analysis based on the experimental data. This resulted in the values $d_1 = -0.0077$, $d_2 = 0.721/\text{MPa}$, $d_3 = 11.9$, and $r = 2056.6$. The calculated values are plotted as solid lines in Figure 16, which also includes the experimental values for comparison. The results showed that Equation (11) can accurately represent the actual situation.

Combining Equations (5), (8), (9), and (11), we can obtain the freeze–thaw damage constitutive model that considers the number of cycles:

$$\sigma_1 = \left(1 - \frac{1}{1 + e^{d_1 N + d_2 \sigma_3 + d_3 - r\varepsilon_1'}}\right)\left[E\varepsilon_1 + \mu(\sigma_2 + \sigma_3)\right] + \frac{1}{1 + e^{d_1 N + d_2 \sigma_3 + d_3 - r\varepsilon_1'}}(c_1 N + c_2 \sigma_3 + c_3) \tag{12}$$

The constitutive model expressed in Equation (12) is composed of nine material parameters, of which five parameters ($E$, $\mu$, $r$, $d_3$, and $c_3$) are considered as the necessary parameters for describing the stress–strain behavior of the rock under uniaxial nonfreeze–thaw tests, which are referred to as basic parameters. The remaining four parameters ($c_1$, $d_1$, $c_2$, and $d_2$) are regarded as freeze–thaw and confining pressure parameters, which respectively represent the effects of freeze–thaw cycles and confining pressure on the residual strength and damage factor.

The basic parameters can be widely adopted by conducting uniaxial tests. On the other hand, the number of confining pressure and strength parameters is relatively small, which can be determined by performing a small number of triaxial and freeze–thaw cycle tests. Therefore, the freeze–thaw damage constitutive model presented in Equation (12) is simple to apply and can be easily popularized.

4.5.3. Validation of Freeze–Thaw Constitutive Model

To verify the rationality of the proposed freeze–thaw damage constitutive model (Equation (12)), experiments were conducted on the Basu granite samples under different confining pressures and numbers of freeze–thaw cycles. The obtained stress–strain curves

(Figure 17) were analyzed using Equation (12) to obtain the material parameters, and the results are presented in Table 4. The calculated stress–strain curves are shown in Figure 17a–e and are compared with the experimental data. The results indicate that the proposed constitutive model is rational and accurately describes the rock's behavior under freeze–thaw cycles.

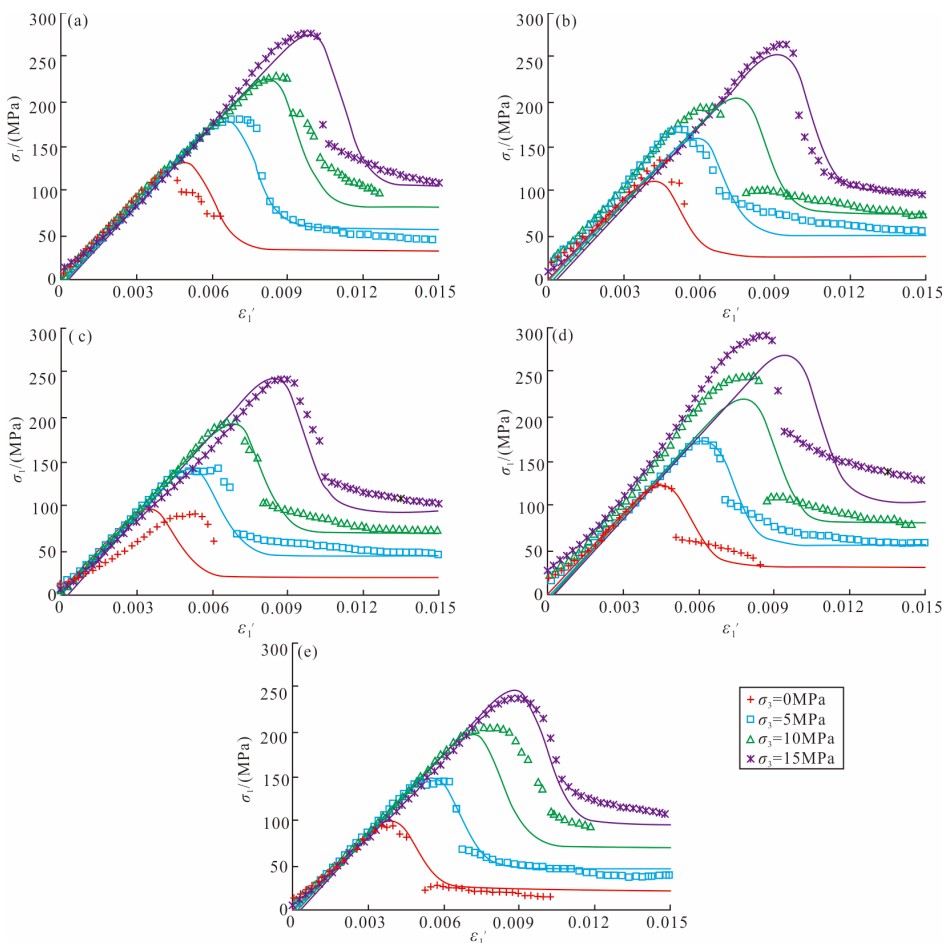

**Figure 17.** The stress–strain curves of the freeze–thaw constitutive model for the Basu granite (points are experimental and full lines are calculated) (**a**) F–T = 0; (**b**) F–T = 10; (**c**) F–T = 20; (**d**) F–T = 30; (**e**) F–T = 40.

**Table 4.** List of parameters for verification of the freeze–thaw constitutive model and their values.

| $E$ (MPa) | $\mu$ | $c_1$ (MPa) | $c_2$ | $c_3$ (MPa) |
|---|---|---|---|---|
| 29,000 | 0.31 | −0.3840 | 4.938 | 33.93 |
| $r$ | $d_1$ | $d_2$ (1/MPa) | $d_3$ | |
| 2056.6 | −0.077 | 0.72 | 11.9 | |

Figure 18a,b depict the impact of freeze–thaw cycles on the stress–strain curve under 0 MPa and 15 MPa confining pressures, respectively, as determined through the constitutive model. The curve shapes are found to be similar for different freeze–thaw cycles. As the number of freeze–thaw cycles increases, the peak strength and corresponding failure strain decrease, along with the residual strength.

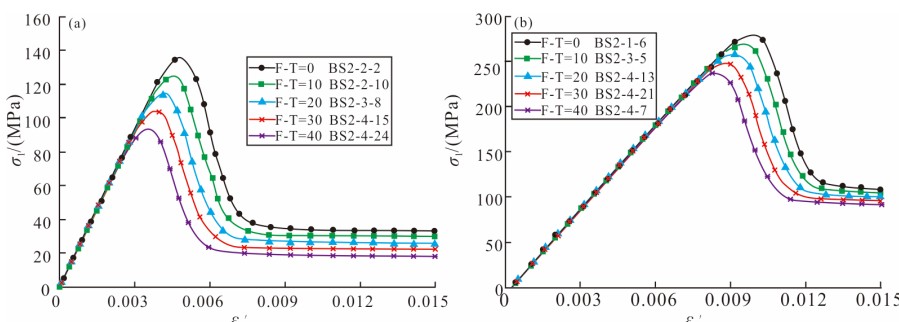

**Figure 18.** Curves of stress–strain with different numbers of freeze–thaw cycles for the Basu granite (**a**) $\sigma_3 = 0$ MPa; (**b**) $\sigma_3 = 15$ MP.

The proposed freeze–thaw damage constitutive model, represented by Equation (12), contains nine material parameters, including five basic parameters, two confining pressure parameters, and two freeze–thaw parameters. The basic parameters can be determined through uniaxial tests, while the confining pressure and freeze–thaw parameters require triaxial and freeze–thaw cycle tests. The simplicity and ease of application of the model make it suitable for a wide range of applications.

### 4.6. Analysis of the Effect of Freeze–Thaw Cycles on the Formation Mechanism of Granite Sand-Sliding Slope Hazards

Sand-sliding slope hazards are a geological phenomenon where the gravel, sand, and debris formed by the weathering of high and steep slopes slide towards the foot of the slope due to gravity, forming a conical slope at the base [10]. These hazards are prevalent in the relatively arid Basu granite mountainous area along the Sichuan–Tibet Road. The sand-sliding slope typically undergoes three stages: the generation of debris and sand particles, sliding, and accumulation at the foot of the slope. Consequently, the sand-sliding slope can be divided into three areas: the sand source area, sliding area, and accumulation area [6]. Two types of sand-sliding slopes have been identified along the Sichuan–Tibet Road G318: naturally growing sand-sliding slopes and those induced by engineering activities [46]. The material composition of the sand-sliding slope in the Basu granite area mainly consists of sand grains with broken stones mixed in, and the material on the slope surface is continually replenished from the upper slope. The slope has high and steep characteristics, making it a growing sand-sliding slope. These sand-sliding slopes are widespread, and due to the uninterrupted sand supply, sand-sliding activities occur frequently, which poses a serious threat to the safety of the road passing through the foot of the slope.

Physical weathering during the formation of sand-sliding slopes is primarily due to temperature changes, and it is known that temperature variation plays a crucial role in the physical weathering of granite and other rocks [24,47,48]. The high altitude of the Basu area of the Tibetan Plateau results in significant temperature differences between the day and night, leading to a freeze–thaw cycle process that causes the long-term expansion and contraction of rocks [10,49]. This paper's study results indicate that the long-term freeze–thaw cycles lead to a single-peak NMR T2 distribution and increased internal cracks in the granite sample, causing the rock to fracture over time. The weakened connection between the surface and interior of the rock results in a lower cohesion and smaller internal friction angle of the granite. The deformation and failure of the rock mass in the granite slope intensifies, eventually loosening it and providing the material basis for the formation of the sand-sliding slope [50,51].

### 5. Conclusions

First, the cracks generated during the freeze–thaw process reduce the integrity of the rock, and the rock's longitudinal wave velocity decreases with increasing freeze–thaw



cycles. The NMR tests revealed that there is only one T2 peak during the freeze–thaw process, and the distribution of the crack volume is relatively concentrated.

Second, the internal friction angle and cohesion of the rock samples after 0, 10, 20, 30, and 40 freeze–thaw cycles were 57.30° and 16.82 MPa; 58.39° and 16.67 MPa; 52.99° and 22.33 MPa; 54.13° and 15.96 MPa; and 54.82° and 14.12 MPa, respectively. The elastic modulus of rock decreases in the range of 20~35 GPa. The internal friction angle and cohesion $c$ and $\varphi$ and the deformation parameter E decrease with the increase in freeze–thaw cycles.

Third, the Poisson's ratio has no obvious variation discipline with the freeze–thaw cycles. The average Poisson's ratio of rock samples was 0.31 and the rock strength increases approximately linearly with increasing confining pressure, which satisfies Mohr–Coulomb's law. However, the strength parameters decrease with increasing freeze–thaw cycles. Weathering cracks are connected by the freeze–thaw cracks, making the rock more broken.

Fourth, based on the damage caused by freeze–thaw cycles to the internal structure of the rock, the authors proposed a freeze–thaw damage constitutive model that considers confining pressure, freeze–thaw cycles, and residual strength. The calculated values were compared with the experimental values, and it was found that the degree of freeze–thaw damage increases as the number of freeze–thaw cycles increases. The peak strength decreases, the corresponding strain decreases, and the residual strength also decreases as the number of freeze–thaw cycles increases.

Fifth, alterations in the internal structural characteristics and strength of granite caused by freeze–thaw cycles are the main factors leading to the occurrence of sand-sliding slopes along the Basu section of the Sichuan–Tibet Road. These slopes are growing sand-sliding slopes.

Overall, this study provides valuable insights into the effects of freeze–thaw cycles on the strength and deformation characteristics of granite and how these changes contribute to the formation of sand-sliding slopes. The proposed freeze–thaw damage constitutive model provides a practical tool for predicting the behavior of rock masses subjected to freeze–thaw cycles. The findings of this study could be of interest to researchers in the fields of geology, geotechnical engineering, and rock mechanics.

**Author Contributions:** C.G. and. J.L. framed the study plan and wrote the paper. J.L. and T.D. conducted data processing and analysis; S.R. participated in the field investigation. Correspondence and requests for materials could be addressed to C.G. (guochangbao@cags.ac.cn). All authors have read and agreed to the published version of the manuscript.

**Funding:** This study was supported by the China Geological Survey projects (Nos. DD20221816, DD20190319).

**Institutional Review Board Statement:** Not applicable.

**Informed Consent Statement:** Not applicable.

**Data Availability Statement:** No new data were created or analyzed in this study. Data sharing is not applicable to this article.

**Acknowledgments:** Thanks to Jiazuo Zhou, the rock mechanical tests were carried out in the Institute of Rock and Soil Mechanics, Chinese Academy of Sciences, China. And thanks to Huimei Zhang from Xi'an University of Science and Technology, China, for the useful rock mechanics experiments discussion. We gratefully acknowledge the four anonymous reviewers and editor for their valuable revision suggestions, which greatly improved the quality of the article.

**Conflicts of Interest:** The authors declare no conflict of interest.

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
