# Peer review of "Investigation of the Deterioration of Basu Granite Mechanical Properties Caused by Freeze–Thaw Cycles in High-Altitude Mountains in the Eastern Part of the Tibetan Plateau, China"

_sustainability, doi:10.3390/su16010319_

Round 1
Reviewer 1 Report
Comments and Suggestions for Authors
Lines 2-4, Title: It is ok and tells about the actual content of the manuscript.
Lines 11-29, Abstract: It is ok too but needs some little corrections concerning the language. I know you report results of test that you have done but normally when you present your contribution it is much better to give verbs in the present simple tense (including its passive voice) and not in the past.
Line 30, Keywords: The provided keywords are ok but to make them more informative you can add extra ones related to methods such as triaxial compressional tests and physical parameters.
Lines 32-78, Introduction: It is well written and shows the problem and the aims of study in a straightforward way.
Line 66, Introduction: If you mean the compressional tests, you should write it in full when it is mentioned for the first time alongside with two brackets, i.e. compressional tests (CT). Then after you can use CT freely.
Line 98: In all places all over the manuscript, please replace "gravel sand" by "gravel and sand".
Line 108, Figure 1: The provided geological maps is ok but it lacks a geographic north (i.e., the N arrow). Also, resolution of the legend would be difficult to produce upon the acceptance of the manuscript, so please either improve or present the map as a separate figure.
Line 129: Please modify caption of Table 1 into: Basic physical indices of the Basu granite samples used in the present study.
Lines 132-133: you write “the” before the “Basu granite”, which is correct. Please apply the same to the rest.
Line 154, Table 2: Only few minor editing is needed.
Line 192, Figure 6: Curves of the T2 distribution of the investigated samples need re-sizing for better resolution and access of infos by the reader.
Line 229, Figure 7: Please indicate what do the SEM images show?. For example, fracturing and what so else.
Line 248, Figure 8: I recommend modification of this figure and separate it into three parts. The first one is devoted to the typical samples then the two curves of the stress-strain below them.
Line 264, Figure 9: Enhance the resolution.
Line 537: Please use “Conclusions” instead of “Conclusion”.
Lines 538-543: Delete this introductory paragraph to the conclusions. You already show them in the introduction, methods and experimental sections. I recommend you summarize your conclusions as short 5-6 bullets.
Lines 687, References: This list is ok and up-to-date. They include all the necessary support for the experimental work by the present authors.
The authors can benefit from the annotated pdf for their revision.

Comments on the Quality of English LanguageMinor language polishing is needed.
Reviewer 2 Report
Comments and Suggestions for Authors
(1) The abstract and conclusion need to include some key quantitative results. In this way, the conclusions drawn from the abstract and conclusion can be effectively supported. Moreover, the rock samples were taken from 3819 kilometers away from the Sichuan Tibet Highway in Basu County (BS1) and the sliding sand slope of the Wangpai Hydropower Station on the left bank of Lengqu (BS2). What is the basis for selecting these two sampling points?
(2) In Table 1, the water content was presented. How was the water content determined in investigation? You should know that if you want to determine the water content, it will inevitably disrupt the frozen state. And what is the significance and purpose of giving the average values of several rock samples in Table1?
(3) In Figure 3a and Figure 4, the horizontal axis represents the rock sample number. I don't think each curve in the graph has much meaning. Because it does not reflect any substantial content. I think the author needs to consider replacing the physical meaning represented by the horizontal axis to make it meaningful again.
(4) In Figure 7, only SEM images under experimental conditions are shown. Only by comparing with the SEM before processing can we better reflect and describe the effect of experimental conditions.
(5) The statement in lines 534-536 needs to be supported by citing some references. https://doi.org/10.1007/s11356-022-21233-7, https://doi.org/10.1007/s11356-022-19663-4.
(6) From Figure 7a, we can see that the growth of peak stress in the rock sample slows down at higher F-T values. What is the reason for this?
(7) We know that the pressure melting effect that occurs under load will more or less affect its mechanical properties. So, should the pressure melting effect be considered when analyzing mechanical properties in this study?
Comments on the Quality of English LanguageSome sentences in paper needs to be rewriten to avoid confusing readers. For example, the sentence in lines 525-527 and 527-530.
Reviewer 3 Report
Comments and Suggestions for Authors
In this paper, mechanical experiments under freeze-thaw cycles were carried out on granite in Basu region and it is well written and readers can benefit from it. The experimental results are useful for revealing the disaster prevention and control of granite in Basu area. Only a few issues need to be addressed in the process of acceptance of the manuscript.
1. The quality of the figures in the manuscript could be enhanced. (figure.1(a), figure.6, figure.8)
2. Whether F-T = 20 in Figure.8 satisfies its stress-strain law?
3. The granite damage pattern in Figure.8(a) can be briefly described, and its corresponding periapsis σ3 can be labeled.
4. Crack sizes in Figure.7 are not clearly labeled, e.g., 31.5μm, etc.
5. How is the friction between the tester indenter and the specimen resolved during triaxial test loading?
6. The manuscript is recommended to be checked by a native English speaker.
Comments on the Quality of English LanguageMinor editing of English language required.
Reviewer 4 Report
Comments and Suggestions for Authors
Round 2
Reviewer 2 Report
Comments and Suggestions for Authors
After the author's revisions, the quality of the manuscript has been greatly improved. At present, its quality has reached the level of recruitment and publication.